# A UNIFIED THEORY OF EARLY VISUAL REPRESENTATIONS FROM RETINA TO CORTEX THROUGH ANATOMICALLY CONSTRAINED DEEP CNNS

**Jack Lindsey**[*][†]**, Samuel A. Ocko**[*]**, Surya Ganguli**[1]**, Stephane Deny**[†]
Department of Applied Physics, Stanford and [1]Google Brain, Mountain View, CA

## ABSTRACT

The vertebrate visual system is hierarchically organized to process visual information in successive stages. Neural representations vary drastically across the first stages of visual processing: at the output of the retina, ganglion cell receptive fields (RFs) exhibit a clear antagonistic center-surround structure, whereas in the primary visual cortex (V1), typical RFs are sharply tuned to a precise orientation. There is currently no unified theory explaining these differences in representations across layers. Here, using a deep convolutional neural network trained on image recognition as a model of the visual system, we show that such differences in representation can emerge as a direct consequence of different neural resource constraints on the retinal and cortical networks, and for the first time we find a *single* model from which *both* geometries spontaneously emerge at the appropriate stages of visual processing. The key constraint is a reduced number of neurons at the retinal output, consistent with the anatomy of the optic nerve as a stringent bottleneck. Second, we find that, for simple downstream cortical networks, visual representations at the retinal output emerge as nonlinear and lossy feature detectors, whereas they emerge as linear and faithful encoders of the visual scene for more complex cortical networks. This result predicts that the retinas of small vertebrates (e.g. salamander, frog) should perform sophisticated nonlinear computations, extracting features directly relevant to behavior, whereas retinas of large animals such as primates should mostly encode the visual scene linearly and respond to a much broader range of stimuli. These predictions could reconcile the two seemingly incompatible views of the retina as either performing feature extraction or efficient coding of natural scenes, by suggesting that all vertebrates lie on a spectrum between these two objectives, depending on the degree of neural resources allocated to their visual system.

## 1 INTRODUCTION

Why did natural selection shape our visual representations to be the way they are? Traditionally, the properties of the early visual system have been explained with theories of *efficient coding*, which are based on the premise that the neural representations are optimal at preserving information about the visual scene, under a set of metabolic constraints such as total firing rate or total number of synapses. These theories can successfully account for the antagonistic center-surround structure of receptive fields (RFs) found in the retina (Atick & Redlich, 1990; 1992; Vincent & Baddeley, 2003; Karklin & Simoncelli, 2011; Doi et al., 2012), as well as for the oriented structure of RFs found in the primary visual cortex V1 (Olshausen & Field, 1996; 1997; Bell & Sejnowski, 1997).

However, a number of properties of the early visual system remain unexplained. First, it is unclear why RF geometries would be so different in the retina and V1. A study (Vincent et al., 2005) has proposed that both representations are optimal at preserving visual information under different metabolic constraints: a constraint on total number of synapses for the retina, and one on total firing rate in V1. However, it is unclear why the two systems would be optimized for these two

---

[*]Equal contribution. All code is available at https://github.com/ganguli-lab/RetinalResources.
[†]Corresponding authors: jackwlindsey@gmail.com and stephane.deny.pro@gmail.com.

different objectives. Second, there is a great diversity of ganglion cell types at the output the retina (Gollisch & Meister, 2010), with each cell type tiling the entire visual field and performing a specific computation. Interestingly, some of these types perform a highly nonlinear computation, extracting specific, behaviorally-relevant cues from the visual scene (e.g. direction-selective cells, object-motion-selective cells), whereas other types are better approximated by a quasi-linear model, and respond to a broad range of stimuli (e.g. midget cells in the primate (Roska & Meister, 2014) and quasi-linear pixel-encoders in the mouse (Johnson et al., 2018)). Intriguingly, although quasi-linear and more nonlinear types exist in species of all sizes (e.g. primate parasol cells are nonlinear (Crook et al., 2008)), the proportion of cells performing a rather linear encoding versus a nonlinear feature detection seems to vary across species. For example, the most common ganglion cell type in the primate retina is fairly well approximated by a quasi-linear pixel-encoder (midget cells, 50% of all cells and >95% in the central retina (Roska & Meister, 2014; Dacey, 2004)), whereas the most common cell type in mouse acts as a specific feature detector, thought to serve as an alarm system for overhead predators (W3 cells, 13% of all ganglion cells (Zhang et al., 2012)). Again, theories of efficient coding have not been able to account for this diversity of computations found across cell types and across species.

The limitations of current efficient coding theories might reside in the simplistic assumption that the objective is to simply relay indiscriminately all visual information to the next stages of processing. Indeed, the ultimate goal of the visual system is to extract *meaningful features* from the visual scene in order to produce an adequate behavioral response, not necessarily to faithfully encode it. A recent line of work has proposed using the information bottleneck framework as a way to move beyond the simplistic objective of information preservation towards more realistic objectives (Chalk et al., 2016; 2018). Another study has shown that by changing the objective from efficiently encoding the present to efficiently encoding the future (predictive coding), one could better account for the spatio-temporal RFs of V1 cells (Singer et al., 2018). Although promising, these approaches were limited to the study of a single layer of neurons, and they did not answer the aforementioned questions about cross-layer or cross-species differences. On the other hand, deep convolutional networks have proven to be accurate models of the visual system, whether they are trained directly on reproducing neural activity (McIntosh et al., 2016; Cadena et al., 2017), or on a behaviorally relevant task (Yamins et al., 2014; Eberhardt et al., 2016; Cadena et al., 2017), but they have not yet been used to study the visual system through the lens of efficient coding theories.

In this study, we trained deep convolutional neural networks on image recognition (CIFAR-10, Krizhevsky (2009)) and varied their architectures to explore the sets of constraints that could have shaped vertebrates' early visual representations through natural selection. We modeled the visual system with a series of two convolutional networks, one corresponding to the retina and one downstream network corresponding to the ventral visual system in the brain. By varying the architecture of these networks, we first found that a reduction in the number of neurons at the retinal output – corresponding to a realistic physical constraint on the number of fibers in the optic nerve – accounted simultaneously for the emergence of center-surround RFs in our model of the retina, and for the emergence of oriented receptive fields in the primary visual relay of the brain. Second, we found that the degree of neural resources allocated to visual cortices in our model drastically reshaped retinal representations. Given a deep visual cortex, the retinal processing emerged as quasi-linear and retained substantial information about the visual scene. In contrast, for a shallow cortex, the retinal processing emerged as nonlinear and more information-lossy, but was better at extracting features relevant to the object classification task. These observations make testable predictions on the qualitative differences that should be found in retinal representations across species, and could reconcile the seemingly incompatible theories of retinal processing as either performing efficient encoding or feature detection.

## 2 FRAMEWORK: A DEEP CONVOLUTIONAL NEURAL NETWORK MODEL OF THE VISUAL SYSTEM

The retinal architecture is strongly conserved across species (Masland, 2001), and consists of three layers of feed-forward convolutional neurons (photoreceptors, bipolar cells, ganglion cells) and two layers of inhibitory interneurons (horizontal, amacrine cells). However, we chose to model the retina as a convolutional neural network (LeCun et al., 2015) with only two layers (fig. 1A). Indeed the retinal response of many species to complex stimuli has been modeled successfully with only one

or two-layer models (Deny et al., 2017; Maheswaranathan et al., 2018; Gollisch & Meister, 2010), with some rare exceptions of models requiring more layers (McIntosh et al., 2016). We refer to this network as the retina-net. In our simulations, we varied the number of neurons in the second layer of the retina-net, which is the output of the retina, corresponding to the physical bottleneck of the optic nerve conveying all the visual information to the brain (fig. 1B).

We modeled the ventral visual system – the system associated with object recognition in the brain (Hubel, 1995) – as a convolutional neural network taking its inputs from the retina-net (fig. 1A). We varied the neural resources allocated to the ventral visual system network (VVS-net) by changing the number of layers it is composed of (fig. 1B).

We trained the neural network composed of the retina-net and VVS-net end-to-end on an object classification task (CIFAR-10, fig. 1A-B-C). Even though the visual system does much more than just classify objects in natural images, this objective is already much more complex and biologically realistic than the one used in previous studies of efficient coding, namely preserving all information about the visual scene. Moreover, we are encouraged by the fact that previous studies using this objective have found a good agreement between neural activity in artificial and biological visual networks (Yamins et al., 2014; Cadena et al., 2017).

More specifically, we trained a convolutional neural network on a grayscale version of the standard CIFAR-10 dataset for image classification. The retina-net consisted of two convolutional layers with 32 channels and $N_{BN}$ channels respectively, and with ReLU nonlinearities at each layer. The VVS-net consisted of a varying number $D_{VVS}$ of convolutional layers with 32 channels followed by two fully connected layers (the first one with 1024 neurons and the second one with 10 neurons mapping to the 10 object categories), with ReLU nonlinearities at each layer and a softmax nonlinearity at the last layer. The full system encompassing the retina-net and VVS-net thus had $32 \rightarrow N_{BN} \rightarrow 32 \rightarrow 32 \rightarrow ...$ channels respectively, where we varied the retinal bottleneck width, $N_{BN}$, as well as the number $D_{VVS}$ of convolutional brain layers (not counting the fully connected layers). In each convolutional layer, we used 9x9 convolutional filters with a stride of 1 at each step. The large filter size was chosen to give the network flexibility in determining the optimal filter arrangement. We trained our network with the RMSProp optimizer for 20 epochs on the training set with batches of size 32. All optimizations were performed using Keras and TensorFlow. For all results presented, we

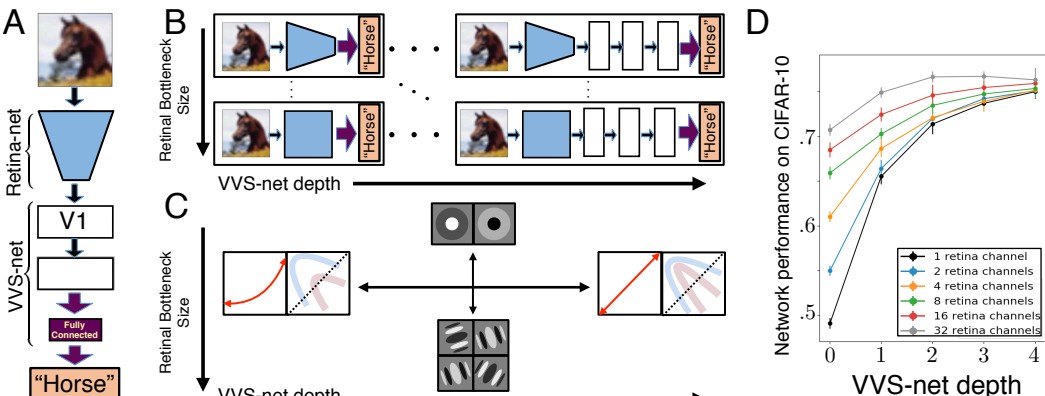

Figure 1: Illustration of the framework we used to model early visual representations. A: We trained convolutional neural networks on an image recognition task (CIFAR-10). The networks were composed of two parts, a retina-net and a ventral-visual-system-net (VVS-net), which receives input from the retina-net. B: We varied the number of layers in the VVS-net (white boxes) and the number of channels at the output of the retina-net (blue box). C: Key results: (1) A bottleneck at the output of the retina yielded center-surround retinal RFs. (2) A shallow VVS-net yielded more nonlinear retinal responses (linearity is schematized by the red arrow), which better disentangled image classes (represented as bent manifolds). D: Test-set accuracy of all model architectures on CIFAR-10, averaged over ten networks with random initial weights for each architecture. Performance increases with VVS-net depth and retinal channel, indicating that both factors are meaningful constraints on the network in the regime tested.

tested statistical significance by training 10 identical networks with different random initializations of weights and biases taken from a Glorot-uniform distribution (Glorot & Bengio, 2010).

After training, we determined the linear approximation of RFs of each convolutional channel of the network in each layer. This was achieved by computing the gradient of the activation of that channel with respect to a blank image. This gradient map gives a first-order approximation of the image pattern that maximally activates the cells in the channel of interest. In the limit of small noise variance, this computation is mathematically equivalent to measuring the cell's spike-triggered average in response to a perturbative white-noise stimulus (Koelling & Nykamp, 2008; Schwartz et al., 2006), a commonly used method for determining receptive fields in experimental biology (Chichilnisky, 2001). This equivalence allowed us to compare directly the geometries of RFs experimentally measured in biological networks with the ones found in our models.

The test accuracy of our neural network model of the visual system at the recognition task increased both with the number of channels in the retinal bottleneck, and with the number of layers in the VVS-net (fig. 1D), confirming that we were in a regime where the restrictions on neural resources in the VVS-net and at the output of the retina were critical to the ability of the network to perform the task.

## 3 A UNIFIED MODEL FOR CENTER-SURROUND RFS IN THE RETINA AND ORIENTED RFS IN V1

Here we investigate the effects of a dimensionality bottleneck at the retinal output on early visual representations in our model of the visual system.

### 3.1 A DIMENSIONALITY BOTTLENECK AT THE RETINAL OUTPUT YIELDS THE EXPECTED REPRESENTATIONS IN RETINA AND V1

When reducing the number of neurons at the output of the retina we found that RFs with antagonistic center and surround emerged. For $N_{BN} = 32$, our control setting with no bottleneck at the retinal output, we observed mostly oriented receptive fields in the second layer of the network (fig. 2A). For $N_{BN} = 4, 2$, and $1$, we observed center-surround receptive fields in the second layer of the network and mostly oriented receptive fields in the third layer, which is the first layer of the ventral visual system in our model (fig. 2B). We quantified these results in App. A. The RF geometries did not depend qualitatively on the VVS-net depth $D_{VVS}$ (results shown for $D_{VVS} = 2$), except for the shallowest VVS-net tested ($D_{VVS} = 0$, no convolutional layer, and thus no dimensionality expansion), for which the shape of emergent retinal RFs were variable across trials and difficult to interpret. These results are in good agreement with the organization of the biological visual system, where retinal RFs are center-surround and most downstream RFs in primary visual cortex (V1) are sharply oriented (Hubel, 1995), suggesting that the dimensionality bottleneck at the output of the retina is sufficient to explain these differences in representations. It is worth noting that for both conditions (bottleneck and no bottleneck), the RFs of downstream layers in the VVS-net after the first layer exhibited complex shapes that were neither clearly oriented, nor circular, and the RFs in the first layer of the retina did not appear to have any well-defined structure (data not shown).

We then tested in our model the hypothesis of Hubel and Wiesel concerning how center-surround cells are pooled to give rise to oriented RFs in V1 (Hubel, 1995). We found that orientation-selective neurons in the VVS-net typically draw primarily from center-surround neurons in the retina-net that are aligned with the direction of the edge, with positive or negative weights corresponding to whether the polarity (light-selective / dark-selective) of the two neurons are consistent or inconsistent (fig. 2C, and App. A for a quantification). These qualitative results are in good agreement with Hubel and Wiesel's hypothesis. Of course, this hypothesis remains to be tested in the real brain, since there is no evidence that the micro-circuitry of the brain matches that of our simulation.

In the visual system of mammals, the main relay of visual information taking its input from the retina is the LGN (thalamus), which has center-surround RFs and a similar total number of neurons as the retinal output (Hubel, 1995). We created a network reflecting this architecture by having two low-dimensionality layers in a row instead of just one (fig. 2C). After training, we found center-surround RFs in the two layers with a bottleneck (retinal output and LGN), and oriented RFs in the next layer, corresponding to the primary visual cortex (V1). These results suggest that center-surround

representations remain advantageous as long as the dimensionality of the representation remains low, and hence dimensionality expansion seems to be the crucial factor explaining the qualitative change of RFs found between LGN and V1.

It is an interesting question to ask whether neurons in our model of the VVS are more similar to simple or complex cells (Hubel, 1995). To test this, we performed a one-step gradient ascent on the neural activity of VVS neurons with respect to the image, starting from several random initial images (App. B). If the neurons were acting as simple cells (i.e. are approximately linear in the stimulus), we would expect all optimized stimuli to converge to the same preferred stimulus. On the other hand, if the cells were complex (i.e. OR function between several preferred stimuli), we would expect the emergent preferred stimuli to depend on the exact initialization. Interestingly, we found that most neurons in the first layer of the VVS-net behaved as simple cells, whereas most neurons in the second layer of the VVS-net behaved as complex cells. Note that in biology, both simple and complex cells are found in V1. These results expose the fact that anatomical regions of visual cortex involve multiple nonlinearities and hence may map onto more than one layer of our simple model. Indeed, V1 itself is a multilayered cortical column, with LGN inputs coming in to layer 4, and layer 4 projecting to layers 2 and 3 (Hubel, 1995). Simple cells are predominantly found in layer 4 and complex cells are predominantly found in layers 2 and 3. These observations bolster the interpretation that biological V1 may correspond to multiple layers in our model.

Local divisive normalization (i.e. local gain control) is an ubiquitous source of nonlinearity in the visual system (Geisler & Albrecht, 1992; Heeger, 1992; Deny et al., 2017). We thus tested the robustness of our main result to a more realistic model of the visual system with local normalization, by adding it at every layer of the network (App. C). We found that receptive fields still emerged as center-surround in the retina-net, and as oriented in our model of V1. We note that the local normalization slightly degraded the performance of the network on the task for all parameter settings we tried.

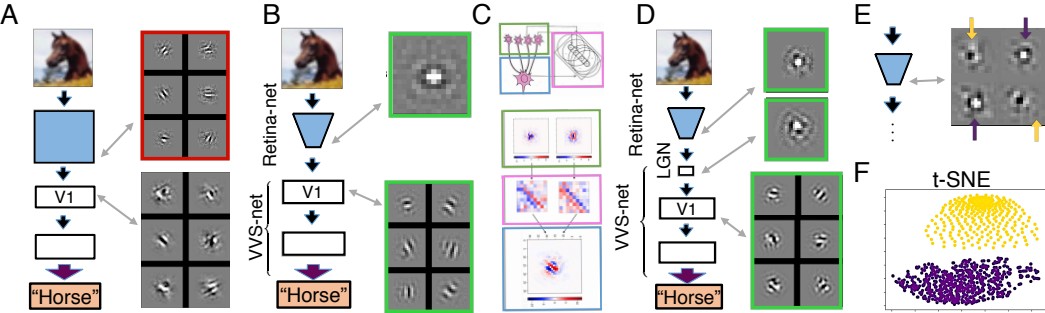

Figure 2: Effects of a bottleneck constraint on receptive fields (RFs). All results are shown for $D_{VVS} = 2$. A: Examples of RFs of cells at selected layers (layers 2 and 3) of a control network with no bottleneck. No center-surround RFs appear. B: Center-surround RFs emerge at the output of the retina-net (layer 2) and oriented RFs emerge in the first layer of the VVS-net when we impose a bottleneck constraint at the output of the retina ($N_{BN} = 1$) C: Top: Hubel and Wiesel's hypothesis on oriented cell formation in V1 (Hubel, 1995). Bottom: A representative example of an orientation-selective neuron (bottom RF) drawing from center-surround channels (top RFs) in the previous layer with weight matrices (center) according to their polarity. Light / dark-selective regions of a receptive field, and positive / negative weights, are represented with red / blue, respectively. D: Examples of RFs in a network with an extra bottleneck layer corresponding to mammalian LGN. Center-surround RFs appear at both the retinal output and LGN layer. E: Examples of ON and OFF center-surround RFs in the untied network ($N_{BN} = 4$). F: t-SNE clustering of the retinal neurons of the untied network (see text). Two distinct cell type clusters form corresponding to ON and OFF center-surround receptive fields.

### 3.2 EMERGENCE OF ON AND OFF POPULATIONS OF CENTER-SURROUND CELLS IN THE RETINA

We then verified that the emergence of center-surround RFs in the retina-net is a consequence of reducing the number of neurons at the retinal output, not of reducing the number of channels, our model's equivalent of biological retinal cell types. In the retina, there exist 20-30 types of ganglion cells (Roska & Meister, 2014), each with a different and stereotyped receptive field, density, polarity (i.e. ON or OFF), and nonlinearities. Cells of each type tile the entire visual field like a convolutional channel in our model, so there is a direct analogy between channels in our model and ganglion cell types in the retina. In order to test whether the emergence of center-surround RFs depends on the number of types that we allow, or just on the number of neurons that we allow at the output of the retina (i.e. dimensionality bottleneck), we employed locally connected layers – equivalent to convolutional layers, but without parameter-tying between artificial neurons within a channel at different spatial locations. In this manner, we can limit the number of neurons at the retinal output without imposing a constraint on the number of cell types. Such a network contains too many parameters to be trained from scratch by gradient descent; to work around this, we trained the model stage-wise by first training our convolutional control network ($N_{BN} = 32$ with parameter tying) and then we trained a three-layers untied network (with bottleneck dimension $N_{BN} = 4$ in the second layer) to reproduce the edge-like activations of the second layer of the control network. Even in the untied retina-net, in which each neuron is effectively its own channel, we found that center-surround RFs emerged (fig. 2E), indicating that center-surround RFs are the network's preferred strategy for passing information through a dimensionality bottleneck even when no constraint on the number of cell types is imposed. We then found that the cells cluster in two distinct populations. To demonstrate this, we measured their activations in response to 10000 natural images, computed the first 20 principal components of this 10000-dimensional space, and ran t-SNE to visualize the clustering of neuron types. We found that two distinct clusters emerged, that corresponded visually to ON and OFF center-surround RFs (fig. 2F). We thus observe in our model the emergence of one of the most prominent axes of dichotomy of biological ganglion cell types, namely the classification of cells in ON and OFF populations with RFs of opposite polarity.

## 4  RETINAL REPRESENTATIONS ARE A FUNCTION OF THE NEURAL RESOURCES ALLOCATED TO THE VENTRAL VISUAL STREAM

To what extent are retinal representations in our model shaped by the degree of neural resources allocated to downstream processing? To investigate this question, we studied the effects of varying the degree of neural resources in the VVS-net, on emergent visual representations in the retina-net.

### 4.1  THE RETINA BECOMES MORE LINEAR AS BRAIN COMPLEXITY INCREASES

As we increased the number of layers in the VVS-net, the retinal computation became more linear (fig. 3A), as measured by the ability of the raw image to linearly map onto the neural representation at the retinal output (see methods, and App. F for a visualization of retinal representation as VVS-net depth increases). This observation is consistent with the current state of knowledge of the differences found in retinal representations across vertebrate species with different brain sizes. The linearization of the retinal response with increased brain complexity was true for different values of bottleneck $N_{BN}$. However, when we did not use any bottleneck ($N_{BN} = 32$), the trend became non-monotonic, with a peak in linearity of the response when the VVS-net had 1 conv layer (data not shown). Another interesting phenomenon to note is that linearity of the retinal response decreased as we increased the number of channels in the bottleneck, at any fixed brain depth (fig. 3A).

The two main sources of nonlinearity in the retina are thought to be the inner retinal rectifications (bipolar and amacrine cells, corresponding to the first rectified layer in our model) and the ganglion cell rectification (corresponding to the second rectified layer in our model). As we decreased VVS-net depth, we observed that the retinal response became more nonlinear. Is this increase in response nonlinearity due to the first or second stage of nonlinearity in our retina-net? To test this, we plotted the real response against the response predicted by a purely linear model for the most shallow and for the deepest VVS-nets tested (fig. 3B). If the linear prediction were inaccurate because of the first stage of nonlinear processing in the retina-net, we would expect the points on the scatter plot to be

scattered around the unit line. If the prediction error were due to the second-stage of nonlinearity, we would expect the linear approximation to make incorrect negative predictions for inactive neurons. In practice, we found that the prediction error of the linear model was partly explained by both stages of nonlinearity in the retina-net model, predicting that both inner retinal nonlinear processing and ganglion cell rectifications should be more pronounced in animals with fewer neural resources in their visual cortices.

## 4.2 THE RETINAL REPRESENTATION IS THE RESULT OF A TRADE-OFF BETWEEN INFORMATION TRANSMISSION AND FEATURE EXTRACTION

Why would retinal representations be more linear when the subsequent ventral visual stream has more resources? One hypothesis is that with a restricted number of neurons, the retina must trade-off between the two incentives of (1) compressing visual information in order to transmit it to downstream layers and (2) extracting nonlinear features from the scene to start disentangling the manifolds corresponding to different classes of objects (Chung et al., 2018a;b). According to this hypothesis, when the VVS is shallow, the priority of the retina should be to work toward extracting relevant features. When the VVS is deep, the priority of the retina should be to transmit as much visual information as possible for downstream processing. We validated this hypothesis in two ways in our model.

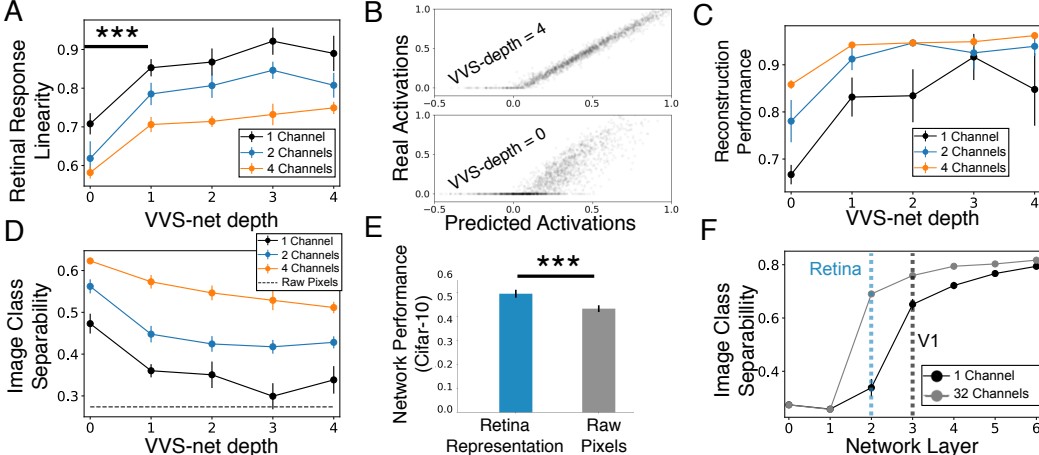

Figure 3: Emergent retinal representations are function of the depth of downstream visual cortices. All error bars represent the 95% confidence interval about the mean (all simulations were repeated over 10 networks trained from random initial conditions). Three stars indicate t-test significance (p<0.001). A: Linearity of the retinal response increases with the number of layers in the VVS-net. Note that it also decreases with the number of cells at the retinal output (different lines). B: Responses of example retina-net output cell to natural images, vs. best linear fit prediction from raw image, for most (top) and least (bottom) deep VVS-nets. Nonlinearity arises from two sources: rectification within the retina-net (corresponds to the spread of the bulk of the point cloud) and rectification at the retina-net output (corresponds to inactive neurons being incorrectly predicted to have negative activations). C: Quality of image reconstruction from the retinal representation as a function of VVS-net depth. The retinal representation retains more information about the raw image for deep VVS-nets. D: Linear separability of classes of objects at the retinal output, as a function of VVS-net depth. Dashed line indicates separability of classes of images from the raw image pixels. Classes are less separable at the retinal output for deeper VVS-nets. E: Performance on CIFAR-10 for a two-layer densely connected network taking its input from the retina-net or from a raw image. Class information is more accessible from retinal representation. F: Class separability at all layers of network for a deep VVS-net ($D_{VVS} = 4$) with and without bottleneck ($N_{BN} = 1$ and $N_{BN} = 32$). Retinal representation of bottleneck network has low separability. However, the first layer of the VVS-net has high separability (see text).

First we showed that the retinal representation retained more information about the image as VVS-net complexity increased (fig. 3C). To estimate information retention, we trained a linear decoder (see methods) from the output of the retina to reconstruct the image and we measured the reconstruction error. The reconstruction error provided a lower bound on the information that the retina retained about the stimulus (note that more information for reconstruction might be accessible by a nonlinear decoder). This result corroborated our hypothesis that, as the VVS-net becomes more complex, the retinal representation gets better at retaining visual information for further processing by the VVS-net.

Second, we found that different classes of objects of CIFAR-10 (e.g. trucks, frogs) were more linearly separable from the retina-net representation when the VVS-net was shallow than when it was deep (fig. 3D). To measure linear separability of manifolds, we trained a linear SVM decoder to separate all pairs of classes and evaluated the performance of the SVM classifier on held-out images (see methods). Moreover, we showed that a VVS-net consisting of two fully connected layers only (no convolutional layers) equipped and trained end-to-end with a retina with a tight bottleneck $N_{BN} = 1$ (dimensionality of retinal output matches dimensionality of the input image) performed better at image recognition than the same VVS-net trained without a retina-net, taking raw images as input (fig. 3E). Both these results corroborate our hypothesis that retinas followed by a simple cortex performs meaningful feature extraction, whereas retinas followed by more complex visual cortices prioritize non-lossy encoding, postponing feature extraction to downstream layers that are better equipped to do it.

Next, we show that within a single network, each retinal channel is trading-off between (1) linearly transmitting visual information to the brain, and (2) extracting relevant features for the object classification task. For 10 instantiations of a network with a retinal bottleneck containing 4 channels, we represented the linearity of each of these 4 channels against the linear separability of object categories obtained from each of these representations. We found, across all networks, a systematic negative correlation between linearity and linear separability across all 4 channels (App. D). Again, this result strongly suggests that extracting features and transmitting visual information are indeed two competing goals shaping representations in our model of the retina.

In the case of the deepest VVS-nets tested, the retinal processing was quasi-linear for the tightest bottleneck (var.expl. = 0.9, $N_{BN} = 1$, fig. 3A). One might take this result to suggest that the retina-net in such models does little more than copy image information. However the very first layer of the VVS-net after the retina disentangled classes (as measured by linear separability) almost as well as the second layer of a VVS-net without a retina (fig. 3F), suggesting that the retinal representation, while only moderately linearly separable itself, is especially transformable into a representation with a high linear separability. This result suggests that even when the retina-net is quasi-linear, it can still participate in extracting relevant features for downstream processing by the brain. The increased separability allowed by the retinal pre-processing for this deep VVS-net could be due to (1) the linear processing or (2) the slightly nonlinear part of the retinal processing (3) a combination of both linear and nonlinear processing. To distinguish between these hypotheses, we replaced the true retinal processing by its best linear approximation, retrained the VVS-net on the output of this linearized retina, and tested whether separability was as high as with the true retinal processing (App. E). We found that the first layer trained on the output of the linearized retinal representation was indeed much more separable than the first layer of the control network (trained directly on natural images) at separating classes of objects, suggesting that the linear operation done by the retina does indeed play a crucial role in making the representation especially separable for subsequent layers.

## 5 METHODS

To estimate the linearity of the response of retinal neurons, we fit a linear model to predict the neural response from the image on 8,000 images. In order to prevent overfitting, we regularized the linear weights with an L2 penalty and optimized the weights using ridge regression. The value of the penalty term was chosen by 10-fold cross-validation on the training set. We then measured the Pearson correlation between the linearized responses and original model responses on a testing set of 2,000 images.

To estimate the information about the input image retained by the retinal output representation, we fit a linear model to reconstruct the image from the (fixed) outputs of the trained retina-net of interest. All numerical figures given are variance-explained results on the held-out test set.

To estimate the linear separability of classes of objects from the neural representation, we trained an SVM classifier between all pairs of classes on half of the testing set of CIFAR-10 (1,000 images that were not used to train the network), and we tested the performance of the SVM classifier on 1,000 held-out images from the testing set, as measured by the percentage of images classified correctly. We then averaged the performance of the SVM across all pairs of classes to obtain the linear separability score.

# 6  DISCUSSION

A unified theoretical account for the structural differences between the receptive field shapes of retinal neurons and V1 neurons has until now been beyond the reach of efficient coding theories. Karklin & Simoncelli (2011) found that efficient encoding of images with added noise and a cost on firing rate produce center-surround RFs, whereas the same task without noise produces edge detectors. However, this observation (as they note) does not explain the discrepancy between retinal and cortical representations. Vincent et al. (2005) propose a different set of constraints for the retina and V1, in which the retina optimizes for a metabolic constraint on total number of synapses, whereas V1 optimizes for a constraint on total firing rate. It is not clear why each of these constraints would predominate in each respective system. Here we show that these two representations can emerge from the requirement to perform a biologically relevant task (extracting object identity from an image) with a bottleneck constraint on the dimensionality of the retinal output. Interestingly, this constraint differs from the ones used previously to account for center-surround RFs (number of synapses or total firing rate). It is worth noting that we unsuccessfully tried to reproduce the result of Karklin & Simoncelli (2011) in our network, by adding noise to the image and applying an L1 regularization to the retina-net activations. In our framework (different than the one of Karklin & Simoncelli (2011) in many ways), the receptive fields of the retina-net without bottleneck remained oriented across the full range of orders of magnitude of noise and L1 regularization that permitted successful task performance.

There is a long-standing debate on whether the role of the retina is to extract relevant features from the environment (Lettvin et al., 1959; Gollisch & Meister, 2010; Roska & Meister, 2014), or to efficiently encode *all* visual information indistinctly (Barlow, 1961; Atick & Redlich, 1990; 1992). In this work, we show that our model of the visual system, trained on the same task and with the same input statistics, can exhibit different retinal representations depending on the degree of neural resources allocated to downstream processing by the ventral visual stream. These results suggest the hypothesis that, despite its conserved structure across evolution, the retina could prioritize different computations in different species. In species with fewer brain resources devoted to visual processing, the retina should nonlinearly extract relevant features from the environment for object recognition, and in species with a more complex ventral visual stream, the retina should prioritize a linear and efficient transmission of visual information for further processing by the brain. Although all species contain a mix of quasi-linear and nonlinear cell types, the proportion of quasi-linear cells seems to vary across species. In the mouse, the most numerous cell type is a two-stage nonlinear feature detector, thought to detect overhead predators (Zhang et al., 2012). In contrast, the most common ganglion cell type in the primate retina is fairly well approximated by a linear filter (midget cells, 50% of all cells and >95% in the central retina (Roska & Meister, 2014; Dacey, 2004)). Note however that two-stage nonlinear models are also present in larger species, such as cat Y-type cells and primate parasol cells (Crook et al., 2008), making it difficult to make definitive statements about inter-species differences in retinal coding. To gain a better understanding of these differences, it would be useful to collect a dataset consisting of recordings of complete populations of ganglion cells of different species in response to a common bank of natural scenes.

A related question is the role of the parcellation of visual information in many ganglion cell types at the retinal output. A recent theory of efficient coding has shown that properties of midget and parasol cells in the primate retina can emerge from the objective of faithfully encoding natural movies with a cost on the total firing rate traversing the optic nerve (Ocko et al., 2018). On the other hand, many cell types seem exquisitely sensitive to behaviorally relevant features, such as

potential prey or predators (Gollisch & Meister, 2010). For example, some cell types in the frog are tuned to detect moving flies or looming predators (Lettvin et al., 1959). It is an intriguing possibility that different cell types could subserve different functions within a single species, namely efficient coding of natural scenes for some types and extraction of behaviorally-relevant features for others. In this study we allowed only a limited number of cell types (i.e. convolutional channels) at the retinal output (1 to 4), in order to have a dimensionality expansion between the retinal representation and the representation in the ventral visual stream (32 channels), an important condition to see the retinal center-surround representation emerge. By using larger networks with more channels in the retina-net and the VVS-net, we could study the emergence of a greater diversity of neuron types in our retina-net and compare their properties to real retinal cell types. It would also be interesting to extend our model to natural movies. Indeed, most feature detectors identified to date seem to process some form of image motion: wide-field, local or differential (Roska & Meister, 2014). Adding a temporal dimension to the model would be necessary to study their emergence.

In conclusion, by studying emergent representations learned by a deep network trained on a biologically relevant task, we found that striking differences in retinal and cortical representations of visual information could be a consequence of the anatomical constraint of transmitting visual information through a low-dimensional communication channel, the optic nerve. Moreover, our computational explorations suggest that the rich diversity of retinal representations found across species could have adaptively co-evolved with the varying sophistication of subsequent processing performed by the ventral visual stream. These insights illustrate how deep neural networks, whose creation was once inspired by the visual system, can now be used to shed light on the constraints and objectives that have driven the evolution of our visual system.

### ACKNOWLEDGMENTS

We would like to thank Lane McIntosh, Niru Maheswaranathan, Aran Nayebi, SueYeon Chung, Vardan Papyan, Nora Brackbill, E.J. Chichilnisky for useful discussions and Stephen Baccus for his comments that greatly improved the manuscript. S.G. thanks the Burroughs-Wellcome, McKnight, James S. McDonnell and Simons foundations for support.

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

APPENDIX

## A    QUANTIFICATION OF RECEPTIVE FIELD ISOTROPY IN RETINA AND V1

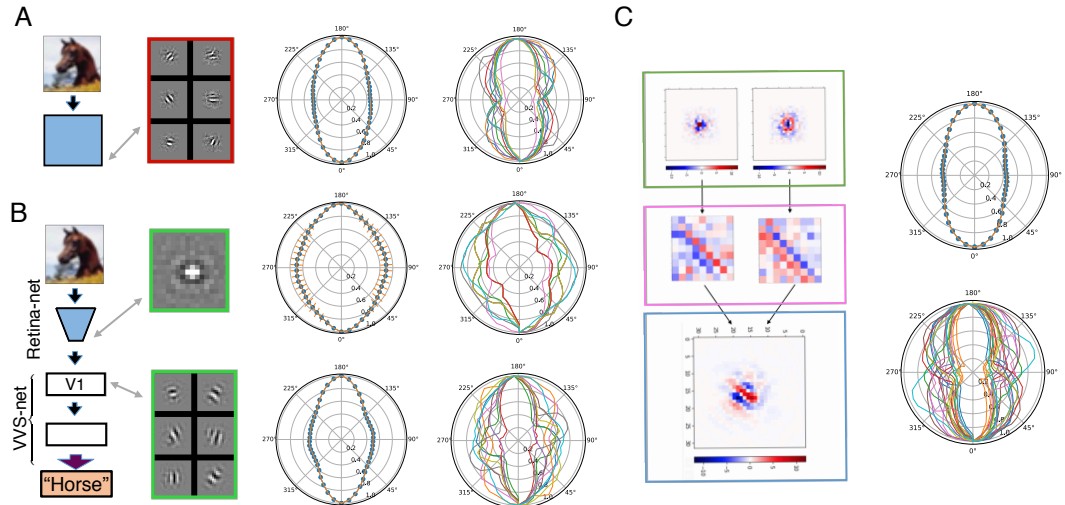

Figure 4: A: Left: Schematic re-illustrating the architecture of a vanilla (no bottleneck) network and showing examples oriented RFs in its second layer. Center: Visualization of average RF isotropy for cells in the second layer of a vanilla convolutional network ($N_{BN} = 1$, $D_{VVS} = 2$). Orange error bars indicate $95\%$ confidence intervals. Right: Visualization of RF isotropy for ten example RFs from the same network architecture. B: Left: Schematic re-illustrating the architecture of the retina-net + VVS-net model ($N_{BN} = 1, D_{VVS} = 2$) and showing example center-surround RFs at the retina-net output and oriented RFs in the following layer (V1). Center and right: Same RF isotropy visualizations as in part A. C: Left: re-illustration of V1 RFs pooling in oriented fashion from center-surround retinal RFs ($N_{BN} = 1, D_{VVS} = 2$). Right: Same isotropy visualizations as in panel A carried on the weight matrix from retina to V1.

The following analysis corroborates our qualitative observation that a dimensionality bottleneck in the retina-net yields center-surround retinal receptive fields and oriented, edge-detecting receptive fields in the first layer of the VVS-net (V1). For a given receptive field, we quantified its orientedness as follows: we displayed rectangular bar stimuli of all possible combinations of width, orientations and spatial translations that fit in the input image window. Among all these combinations, we selected the bar stimulus width, orientation, and translation that yielded the strongest response from the RF. Bars with the same width as the best stimuli were presented at all orientations and translations, and for each orientation, we select the strongest response it produced (across all translations). In this manner we obtained a measure of the strength of a receptive field's preference for all orientations.

We measured the strength of each RF preference (maximum strength of response) for its preferred orientation and for the orthogonal orientation, and computed the ratio of these strengths. Completely isotropic filters would be expected to give a ratio of 1, while oriented filters should give higher ratios. Note however that some deviation from 1 may indicate noise in the filter rather than true orientedness. For each network layer, we averaged this ratio across filters (for convolutional layers with multiple layers) and trials (re-training of the same neural network architecture with different random initializations). We found that the average ratios were $1.56(\pm0.22)$ for the retinal output, $3.05(\pm0.30)$ for the first VVS-net layer, and $2.57(\pm0.27)$ for the second VVS-net layer, where error margins given are $95\%$ confidence intervals. To help assess whether retinal RFs were more isotropic than expected by chance, we compared them to receptive fields composed of random Gaussian noise as a baseline. These give an average ratio (as computed above) of $1.97(\pm0.08)$, significantly higher than that for retinal RFs. Furthermore, the standard deviation of RF preference across orientations was significantly lower for the retinal RFs $(0.118 \pm 0.036)$ than for random RFs $(0.177 \pm 0.007)$, also indicating that retinal RFs were more isotropic than expected by chance.

We also plot the average RF preference for different orientations at each layer to more comprehensively assess the isotropy of RFs at each network layer. To aggregate results across multiple trials and filters, we rotated the coordinates of each receptive field such that its preferred orientation was vertical, and averaged our results across filters and trials. (See Figure 4).

The results confirm our qualitative observations that (1) RFs in the second layer of a vanilla network ($N_{BN} = 32$) are highly oriented (Figure 4A) (2) RFs in the second layer (retina output) of a bottleneck network ($N_{BN} = 1$) are much more isotropic, consistent with center-surround RFs (Figure 4B top), and (3) RFs in the layer immediately following the retina-net in the bottleneck network are oriented (Figure 4B bottom).

We also quantitatively corroborate our observation that oriented receptive fields in the V1 layer pool input from oriented arrays of center-surround filters in the retina-net output layer. We apply our method of isotropy quantification described above to the weight matrix for each input-output filter combination in the V1 convolutional layer. We find that this weight matrix itself exhibits orientedness across filters and trials, confirming our observation (Figure 4C).

## B  SIMPLE AND COMPLEX CELLS

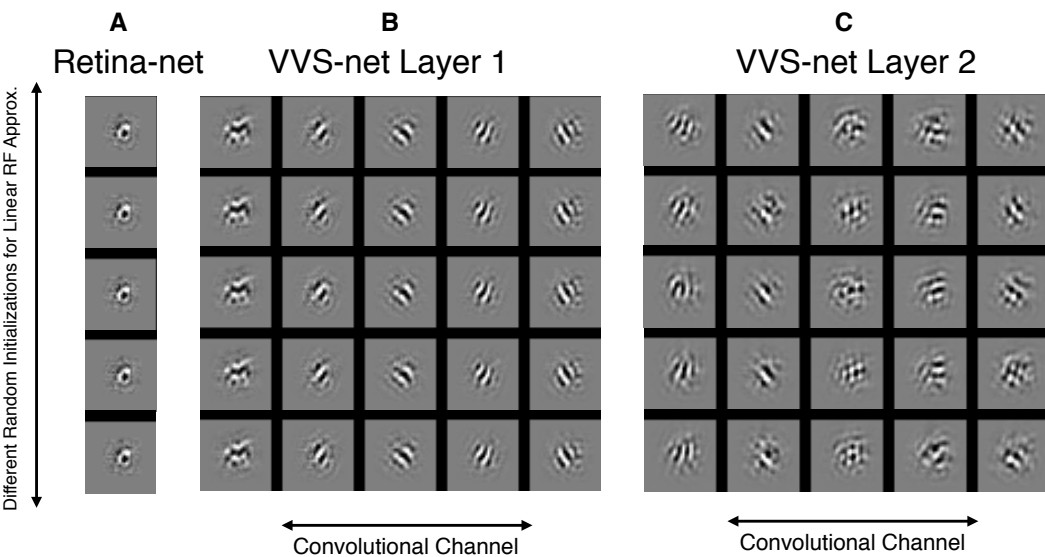

Figure 5: A: Visualizations of retina-net output RFs for an example network ($N_{BN} = 1, D_{VVS} = 2$) using different random initialization, as described in the text. B: Same as A, for the first layer of the VVS-net, and showing 5 of the layer's 32 channels on the x axis. C: Same as B, for the second layer of the VVS-net. In contrast to the first layer, the emergent preferred stimuli are always different across different initializations, indicative of a complex-cell like behavior.

To investigate whether neurons in our model's early layers more closely resembled simple or complex cells, we performed the following analysis. As before, we obtained local linear approximations of receptive fields by computing the gradient in input space with respect to the response of a given neuron. Rather than beginning with a blank input, we ran multiple trials with different randomly initialized inputs. A purely linear cell would give the same result no matter the initialization; a somewhat nonlinear but still "simple" cell is expected to give similar results across initializations. A "complex" cell is expected to give different RF visualizations for different random inputs, reflecting multiple peaks in its response as a function of input. In Figure 5 we show examples of receptive fields at different layers of our retina-net + VVS-net model (with $N_{BN} = 1, D_{VVS} = 2$) for different random intializations of the image (uniform random in [0, 1]). The retina-net output and first VVS-net layer exhibit "simple" behavior, but the second VVS-net layer exhibits observably "complex" behavior. To quantify this effect, we measure the average (across filters within each layer and re-trainings of the same network architecture) standard deviation of computed RFs (normalized to the range [0, 1]) for each network layer. We found that the average standard deviations

were $7.9(\pm 1.1) \times 10^{-3}$, $15.4(\pm 0.8) \times 10^{-3}$, and $35.9(\pm 0.8) \times 10^{-3}$ for the retina-net output, first VVS-net layer, and second VVS-net layer, respectively, where the margins of error given are 95% confidence intervals. These results corroborate the observation of significantly more complex behavior in the second VVS-net layer, mirroring the biological phenomenon in which complex cells pool from simple cells in V1.

## C  EFFECTS OF LOCAL RESPONSE NORMALIZATION ON EARLY VISUAL REPRESENTATIONS

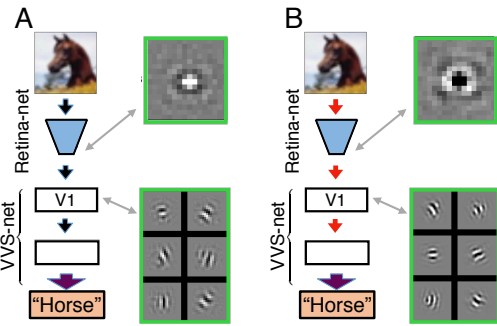

Figure 6: Example RFs from the bottleneck network ($N_{BN} = 1, D_{VVS} = 2$) without (A) and with (B) local response normalization (i.e. local gain control).

We tested the robustness of our first main finding – that a bottlenecked retina-net + VVS-net model yields center-surround receptive fields in the retina and oriented receptive felds in V1 – to the use of biologically realistic local response normalization at every layer of the network. In particular, we normalized the output $x$ of each channel (row $r$, column $c$) of each layer as follows (during training and testing):

$$x_{r,c} \leftarrow \frac{x_{r,c}}{\left(k + \alpha \sum_{r' \in [r-\frac{n}{2}, r+\frac{n}{2}], c' \in [c-\frac{n}{2}, c+\frac{n}{2}]} x_{r',c'}\right)^{\beta}}$$

where the subscripts of $x$ indicate the spatial location (row/column), and $k$, $\alpha$, ad $\beta$ are constants. We used $k = 2$, $\beta = 0.5$ and $\beta = 0.75$, and $\alpha = 5 \times 10^{-4}$ and $\alpha = 5.0$. All parameter settings tested yielded RFs with the same qualitative properties as in the model without normalization. Figure 6 shows example RFs from the no-normalzation model next to example RFs from the normalization model with $k = 2$, $\beta = 0.5$, $\alpha = 5.0$.

## D   RETINAL CELL TYPES TRADE OFF BETWEEN LINEAR INFORMATION TRANSMISSION AND NONLINEAR FEATURE EXTRACTION

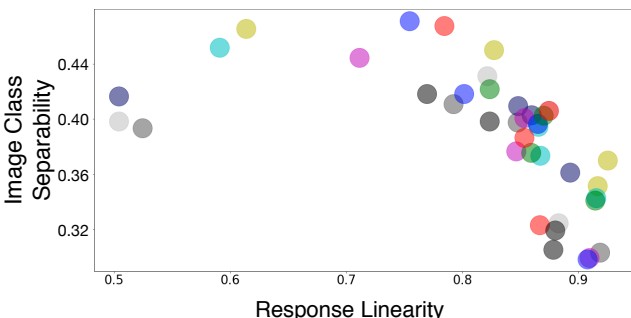

Figure 7: Linearity vs. Class separability for each retina-net output channels (i.e. bottleneck layer). VVS-Net depth is equal to 4. Each network has a bottleneck size of 4 channels (i.e. $N_{BN}$ =4). Distributions are plotted across 10 network instances; each point represents a single channel, colored according to its network. The negative slope suggests that there is trade-off between linearly transmitting visual information for downstream processing (i.e. efficient coding) and extracting useful features for the object recognition task.

## E   LINEARIZED RETINA ALSO INCREASES SEPARABILITY IN SUBSEQUENT LAYERS

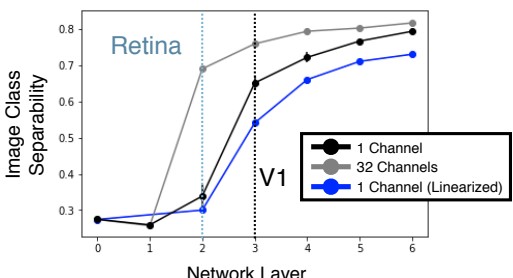

Figure 8: Class separability at all layers of network for a deep VVS-net ($D_{VVS}$ = 4) with and without bottleneck ($N_{BN}$ = 1 and $N_{BN}$ = 32). Retinal representation of bottleneck network has low separability. However, the first layer of the VVS-net has high separability. We additionally plot the separability of the linearized bottleneck ($N_{BN}$ = 1) network (see test) as a function of layer. That the jump in linear separability between layers 2,3 survives linearization suggests that the main effect of retinal processing in this network is whitening (see Fig. 9) rather than nonlinear processing.

In the case of the deepest VVS-nets tested, the retinal processing was quasi-linear for the tightest bottleneck (var.expl. = 0.9, $N_{BN} = 1$, fig. 3A). However the very first layer of the VVS-net after the retina disentangled classes (as measured by linear separability) almost as well as the second layer of a VVS-net without retina (fig. 3F), suggesting that the retinal representation, while only moderately linearly separable itself, is especially transformable into a representation with a high linear separability. To determine to what degree this increased separability was due to (1) the linear processing or (2) the slightly nonlinear part of the retinal processing, we performed an ablation experiment to eliminate factor (2). We first replaced the true retinal processing by its best approximation by a one-layer linear convolution (of sufficient filter width to correspond to two convolutional layers with 9 by 9 filters). After this linearization process, we retrained the VVS-net using the *linearized* retinal representation as input, keeping the linearized retina weights frozen. We found that the first layer trained on the output of the linearized retinal representation was indeed much better than the first layer of the control network (trained directly on natural images) at separating classes of objects (Fig.

8), suggesting that the linear operation done by the retina does indeed play a crucial role in making the representation especially separable for subsequent layers. Visualization of retinal processing in App. F suggest that whitening is an important part of this linear processing.

# F    RETINAL REPRESENTATION VISUALIZATION AS A FUNCTION OF VVS-NET DEPTH FOR BOTTLENECK $N_{BN} = 1$

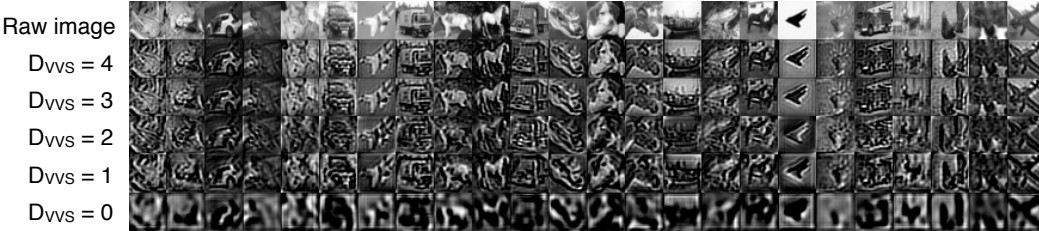

Figure 9: Visualization of the output of the retina-net (one-channel-bottleneck, i.e. $N_{BN} = 1$) for different images from the testing set (x-axis) as a function of VVS-net depth (y-axis). Each pixel intensity of the retinal image is proportional to the activation of the corresponding neuron of the retina, where light shades indicate high activities and dark shades low activities. While retinas for every VVS-net depth appear to whiten the input, we can see that the retinal image is more and more processed and less and less recognizable as VVS-net depth decreases.

