# OpenReview forum: "A Unified Theory of Early Visual Representations from Retina to Cortex through Anatomically Constrained Deep CNNs"
_ICLR.cc/2019/Conference_

### Official Review · AnonReviewer1 · 2018-10-29
**Interesting application of deep neural nets to neuroscience.**

**Rating:** 8
**Confidence:** 3

**Review:**

This paper addresses questions about the representation of visual information in the retina. The authors create a deep neural network model of the visual system in which a single parameter (bandwidth between the “retina” and “visual cortex” parts) is sufficient to qualitatively reproduce retinal receptive fields observed across animals with different brain sizes, which have been hard to reconcile in the past.

This work is an innovative application of deep neural networks to a long-standing question in visual neuroscience. While I have some questions about the analyses and conclusions, I think that the paper is interesting and of high quality.

My main concern is that the authors only show single examples, without quantification, for some main results (RF structure). For example, for Fig. 2A and 2B, an orientation selectivity index should be shown for all neurons. A similar population analysis should be devised for Fig 2C, e.g. like Fig 3 in [1]

Minor comments:
1. Page 4: “These results suggest that the key constraint ... might be the dimensionality bottleneck..”: The analyses only show that the bottleneck is *sufficient* to explain the differences, but “the key constraint” also implies *necessity*. Either soften the claim or provide control experiments showing that alternative hypotheses (constraint on firing rate etc.) cannot explain this result in your model.

2. I don’t understand most of the arguments about “cell types” (e.g. Fig. 2F and elsewhere). In neuroscience, “cell types” usually refers to cells with completely different connectivity constraints, e.g. excitatory vs. inhibitory cells or somatostatin vs. parvalbumin cells. But you refer to different CNN channels as different “types”. This seems very different than the neuroscience definition. CNN channels just represent different feature maps, i.e. different receptive field shapes, but not fundamentally different connectivity patterns. Therefore, I also don’t quite understand what you are trying to show with the weight-untying experiments (Fig. 2E/F).

3. It is not clear to me what Fig. 3B and the associated paragraph are trying to show. What are the implications of the nonlinearity being due to the first or second stage?

4. Comment on Fig 3F: The center-surround RFs probably implement a whitening transform (which is linear). Whitened inputs can probably be represented more efficiently in a network trained with L2-regularization and/or SGD. This might explain why the “quasi-linear” retina improves separability later-on.

[1] Cossell, Lee, Maria Florencia Iacaruso, Dylan R. Muir, Rachael Houlton, Elie N. Sader, Ho Ko, Sonja B. Hofer, and Thomas D. Mrsic-Flogel. “Functional Organization of Excitatory Synaptic Strength in Primary Visual Cortex.” Nature 518, no. 7539 (February 19, 2015): 399–403. https://doi.org/10.1038/nature14182.

---

> ### Author Response · Authors · 2018-11-16
> **Thank you for thoughtful suggestions - Clarity and quantification concerns adressed**
>
> We thank the reviewer for their positive appreciation, and for their thoughtful suggestions that we took into account.
>
> [Main concern - Quantifications for Fig 2A,B and C]
> Fig2A and 2B: We quantified our result about the isotropy of retinal filters, by measuring orientedness of the filters at the retinal output and in V1 on 10 different instantiations of the network. We show that the retinal filters are significantly more isotropic than RFs in both the control network (without bottleneck) and the V1 filters.
> Fig 2C (Hubel and Wiesel hypothesis): We quantified the anisotropy of the weight filter pooling from the retina to form oriented filters in V1, and again we found that these weight matrices are significantly oriented, confirming the hypothesis of Hubel and Wiesel in our model that simple cells in V1 are built from pooling successive center-surround filters from the preceding layer in a row. These quantifications are now referred to in the main text and detailed in the appendix.
>
> [1. Bottleneck is a sufficient constraint, not a necessary constraint]
> We agree with the reviewer that we cannot eliminate other hypotheses about the origin of isotropic filters in the biological retina. We soften the claim everywhere in the manuscript as suggested. Note that we mention in the discussion our attempt to reproduce the results of Karklin and Simoncelli (2011), who could successfully obtain center-surround RFs with a constraint on firing rate, but with a different objective function (information preservation). However, we cannot totally eliminate the possibility that, with different network parameters, we could also obtain center-surround Rfs with a constraint on total firing rate under this object recognition objective.
>
> [2. Cell types in the retina]
> We understand the confusion of the reviewer and we clarify this point both here and in the manuscript. The retina is organized in layers of different types of neurons (photoreceptors, bipolar cells, ganglion cells, etc), and in each of these layers the neurons can be subdivided in many subtypes: these subtypes we referred to as types in the article, which might have led to the confusion. For instance in the primate retina, there exist 20 subtypes of ganglion cells, each with different receptive field size, polarity, non-linearities etc (Dacey 2004). Each of these subtypes tile the entire visual field like a convolutional channel in a layer of a CNN and each cell of a given subtype has a stereotyped receptive field, so this is why there is a strong analogy between channels in our model and biological subtypes. We wanted to test whether the emergence of center-surround RFs in the retina was a consequence of reducing the number of channels (i.e. subtypes), or the number of neurons, and this is why we carried out the experiment described in the section “Emergence of ON and OFF populations of center-surround cells in the retina” where we untied the weights of the network. We find that the emergence of center-surround is not specifically dependent on the number of types that we allow (it only depends on the number of cells that we allow), and furthermore we find that the cells naturally arrange in two clusters of ON and OFF cells when we allow them to differentiate, which is an interesting side-observation because the polarity axis is the first axis of ganglion cell subtype classification in the retina.
>
> [3. implications of the nonlinearity being due to the first or second stage]
> This analysis was directed at retinal experts who might want to test our predictions, and that might wonder what stage of the linear processing is responsible for the non-linearity of the retinal response as we decrease neural resources allocated to the brain. The two main sources of non-linearity in the retina are thought to be the inner retina rectification (bipolar and amacrine cells, corresponding to the first stage non-linearity in our model) and the ganglion cell rectification (corresponding to the second stage non-linearity in our model). We find that both stages become more non-linear as we decrease brain resource, which makes an interesting prediction for experimentalists. We clarified the motivation for this analysis and the corresponding prediction that it makes in the manuscript.

---

> > ### Author Response · Authors · 2018-11-16
> > **Continued**
> >
> > [4. Whitened inputs can probably be represented more efficiently in a network trained with L2-regularization and/or SGD]
> > We thank the reviewer for this interesting explanation that we could directly verify in our model. In the case of a deep brain network, where the retinal processing is quasi-linear, the increased separability allowed by the retinal pre-processing could be due to (1) the linear whitening or (2) the slightly non-linear part of the retinal response (3) a combination of both linear and non-linear processing. To distinguish between these hypotheses, we replaced in a new experiment the true retinal processing by its best linear approximation, retrained the brain network on the output of this linearized retina and tested whether separability was as good as with the true retinal processing. We found that the first layer trained on the output of the linearized retinal representation was indeed much better than the first layer of the control network (trained directly on natural images) at separating classes of objects, suggesting that the linear whitening operation done by the retina is indeed especially transformable into linearly separable representations by a downstream neural network. We added this analysis in the appendix.

---

### Official Review · AnonReviewer3 · 2018-11-02
**A great paper and a solid contribution to computational neuroscience**

**Rating:** 8
**Confidence:** 5

**Review:**

I enjoyed reading this paper which is a great example of solid computational neuroscience work.

The authors trained CNNs under various biologically-motivated constraints (e.g., varying the number of units in the layers corresponding to the retina output to account for the bottleneck happening at the level of the optic nerve or varying the number of "cortical" layers to account for differences across organisms). The paper is clear, the hypotheses clearly formulated and the results are sound. The implications of the study are quite interesting suggesting that the lack of orientation selectivity in the retina would arise because of the bottleneck at the level of the optic nerve. The continuum in terms of degree of linearity/non-linearity observed across organisms at the level of the retina would arise as a byproduct of the complexity/depth of subsequent processing stages. While these results are somewhat expected this is to my knowledge the first time that it is shown empirically in an integrated computational model.

Minor point: The authors should consider citing the work by Eberhardt et al (2016) which has shown that the exists an optimal depth for CNNs to predicting human category decisions during rapid visual categorization.

S. Eberhardt, J. Cader & T. Serre. How deep is the feature analysis underlying rapid visual categorization? Neural Information Processing Systems, 2016.

---

> ### Author Response · Authors · 2018-11-16
> **Thanks**
>
> We thank the reviewer for their positive assessment. We agree that some of these observations could be expected but it is the first time to our knowledge that cross-layer and cross-species differences in early visual representations are recapitulated and accounted for in a single unified model of the visual system.
>
> We thank the reviewer for this interesting reference that we added as an example of how deep networks can be used to model the human visual system.

---

### Official Review · AnonReviewer2 · 2018-11-05
**Review of The effects of neural resource constraints on early visual representations**

**Rating:** 8
**Confidence:** 5

**Review:**

EDIT: On the basis of revisions made to the paper, which significantly augment the results, the authors note: "the call for papers explicitly mentions applications in neuroscience as within the scope of the conference" which clarifies my other concern. For both of these reasons, I have changed my prior rating.

This paper is focused on a model of early visual representation in recognition tasks drawing motivation from neuroscience. Overall the paper is an interesting read and reasonably well written (albeit with some typos). The following addresses the positives and negatives I see associated with this work:

Positives:
- There are relatively few efforts that focus heavily on more shallow models with an emphasis on representation learning, and for this reason this paper fills an important space
- The connections to neuroscience are interesting albeit it's unclear the extent to which this is the mandate of the conference
- The most interesting bit of the paper to me is the following: "A bottleneck at the output of the retina yielded center-surround retinal RFs" - it is somewhat a foregone conclusion that most networks immediately converge on orientation selective and color opponent representations. That this model produces isotropic filters is a very interesting point.

Negatives:
- The work feels a little bit shallow. It would have been nice to see a bit more density in terms of results and ablation studies. This also relates to my second point.
- Given the focus on early visual processing, there seems to be a missed opportunity in examining the role of normalization mechanisms or the distinction between simple and complex cells. If the focus resides in the realm of neuroscience and early visual representation, there is an important role to these mechanisms. e.g. consider the degree of connectivity running from V1 to LGN vs. LGN to V1.

---

> ### Author Response · Authors · 2018-11-16
> **Complementary analyses [1]**
>
> We thank the reviewer for the positive comments about the paper, and try to address his/her concerns below.
>
> [Role of normalization mechanisms.]
> Local normalization is an ubiquitous source of non-linearity in the visual system (see Geisler and D.G. Albrecht 1992 for an example in the cortex, and Deny et al 2017 for an example in the retina), and in ML they are used to enhance contrast of images (Lyu and Simoncelli 2008, http://www.cns.nyu.edu/pub/lcv/lyu08b.pdf) and image compression algorithms (Balle et al, ICLR 2017 https://openreview.net/forum?id=rJxdQ3jeg). We thus tested the robustness of our main results to a more realistic model of the visual system with local normalization by adding local normalization at every layer of the network. We found that receptive fields still emerge as center-surround in the retina-net and as oriented in our model of V1 when we put a bottleneck. Interestingly, the normalization slightly degraded the performance of the network on the task for all parameter settings we tried.  We added this complementary analysis in the main text and appendix of the article. (see section 3.1 and App C)
>
> [Distinction between simple and complex cells.]
> It is an interesting question to ask whether neurons in our model of the VVS are more similar to simple or complex cells. To test this, we performed a one-step gradient ascent on the neural activity of VVS neurons with respect to the image, starting from several random initial images. If the neurons were acting as simple cells (i.e. are approximately linear in the stimulus), we would expect all optimized stimuli to converge to the same preferred stimulus. On the other hand, if the cells were complex (i.e. OR function between several preferred stimuli), we would expect the emergent preferred stimuli to depend on the exact initialization. Interestingly, we found that most neurons in the first layer of the VVS-net behaved a simple cells, whereas most neurons in the second layer of the VVS-net behaved as complex cells. Note that in biology, both simple and complex cells are found in V1. These results expose the fact that anatomical regions of visual cortex involve multiple nonlinearities and hence may map onto more than one layer of our simple model. Indeed, V1 itself is a multilayered cortical column, with LGN inputs coming in to layer 4, and layer 4 projecting to layers 2 and 3.  Simple cells are predominantly found in layer 4 and complex cells are predominantly found in layers 2 and 3. These observations bolster the interpretation that biological V1 may correspond to multiple layers in our model. We added these interesting results and observations in the main text and appendix. (see section 3.1 and App B)
>
> [Role of the thalamo-cortical loop.]
> The recurrence of the thalamo-cortical loop also plays an essential role in the computations of the visual system and it would be very important to understand the role of this recurrence. However, in this study we chose to focus on explaining the discrepancy between the geometry of RFs in the retina and V1, and on the differences in the non-linearity of retinal processing across species. To model these phenomena, our approach was to find the simplest model that would yield those two phenomena. Intriguingly, our results show that modeling the thalamo-cortical loop is not necessary to yield the emergence of center-surround receptive fields in the retina-net and oriented receptive fields in the V1 layer (first layer of VVS-net).  Moreover, note that a number of studies of the visual system using those same simplifying assumptions (simple neurons, no recurrence, Yamins et al 2014, Cadena et al. 2017) have found good agreement of the predictions of their models with the visual system.  Also, almost all classical efficient coding theories going back to Atick and Redlich and Olshausen and Field assume no top-down feedback, so it is important to first compare to them using a model without top-down feedback as a first step.

---

> > ### Author Response · Authors · 2018-11-16
> > **Complementary analyses [2]**
> >
> > [More density in terms of results and ablation studies.]
> > We have now quantified our main result about the isotropy of retinal filters, by measuring isotropy of the filters at the retinal output and in V1, on 10 different instantiations of our network. We show that the retinal filters are significantly more isotropic in the network with bottleneck than in the control network without bottleneck. We also find that the filters in V1 are significantly more oriented than in the retina-net. We added these quantifications in the appendix. (see App A)
> >
> > Ablation study: Following a suggestion of Rev. 1, we investigated in depth whether, in the case of a deep brain network, where the retinal processing is quasi-linear, the increased separability allowed by the retinal pre-processing is due to the linear (whitening) or non-linear aspects of the retinal pre-processing (fig 3F). To test this we replaced the actual retinal processing by its best linear approximation (i.e. this is a functional ablation). We then retrained the brain network on the output of this linearized retina and tested whether separability was as good as with the real slightly non-linear retinal processing. We found that separability in the very first layer of the VVS-net was already much stronger than that of a VVS-net trained directly on natural images (without retina). This result demonstrates that linear whitening does indeed play a crucial role in making the representation easily transformable by subsequent layers into a linearly separable representation. We added this analysis in the amin text and appendix. (see section 4.2 and App E)
> >
> > Finally, we did a new analysis suggesting even more strongly that the retinal representation is indeed a trade-off between feature extraction and linear transmission of visual information. For 10 instantiations of a network with a retinal bottleneck containing 4 channels, we represented the linearity of each of these 4 channels against the linear separability of object categories obtained from these representations. We found, across all networks, a systematic negative correlation between linearity and linear separability across all 4 channels. This result strongly suggests that extracting features and  transmitting visual information are indeed two competing goals shaping retinal representations. We added these new results in the appendix. (see section 4.2 and App D)
> >
> >
> > In summary, we have added 5 new complementary analyses to the article, making it substantially denser in terms of both results and ablation studies.

---

### Meta-Review · Area_Chair1 · 2018-12-12
**a good step in bringing computational neuroscience and CNNs together**

**Confidence:** 5
**Recommendation:** Accept (Oral)

**Metareview:**

The paper advocates neuroscience-based V1 models to adapt CNNs.  The results of the simulations are convincing from a neuroscience-perspective.  The reviewers equivocally recommend publication.